# Hidden in Plain Sight: Subgroup Shifts Escape OOD Detection

**Lisa M. Koch**[1]                                              LISA.KOCH@UNI-TUEBINGEN.DE
[1] *Institute for Ophthalmic Research, University of Tübingen, Germany*

**Christian M. Schürch**[2]                  CHRISTIAN.SCHUERCH@MED.UNI-TUEBINGEN.DE
[2] *Department of Pathology and Neuropathology, University Hospital and Comprehensive Cancer Tübingen, Germany*

**Arthur Gretton**[3]                                              ARTHUR.GRETTON@GMAIL.COM
[3] *Gatsby Computational Neuroscience Unit, University College London, United Kingdom*

**Philipp Berens**[1]                                        PHILIPP.BERENS@UNI-TUEBINGEN.DE

**Editors:** Under Review for MIDL 2022

## Abstract

The safe application of machine learning systems in healthcare relies on valid performance claims. Such claims are typically established in a clinical validation setting designed to be as close as possible to the intended use, but inadvertent domain or population shifts remain a fundamental problem. In particular, subgroups may be differently represented in the data distribution in the validation compared to the application setting. For example, algorithms trained on population cohort data spanning all age groups may be predominantly applied in elderly people. While these data are not "out-of distribution", changes in the prevalence of different subgroups may have considerable impact on algorithm performance or will at least render original performance claims invalid. Both are serious problems for safely deploying machine learning systems. In this paper, we demonstrate the fundamental limitations of individual example out-of-distribution detection for such scenarios, and show that subgroup shifts can be detected on a population-level instead. We formulate population-level shift detection in the framework of statistical hypothesis testing and show that recent state-of-the-art statistical tests can be effectively applied to subgroup shift detection in a synthetic scenario as well as real histopathology images.

**Keywords:** safety, domain shift detection, subgroups, hypothesis testing, kernel methods

## 1. Introduction

Machine learning (ML) tools for medical image analysis have been approaching human-level performance in controlled settings in various application areas such as ophthalmology, breast, skin and lung cancer detection, respiratory diseases and orthopaedics (Liu et al., 2019). However, major hurdles still obstruct the wide adoption of machine learning in clinical practice. When ML is applied in a clinical setting, its outputs are typically used to ultimately inform treatment decisions. Therefore, as a flipside to their vast potential, ML algorithms can also indirectly cause harm to the patient. For example, failure to detect the

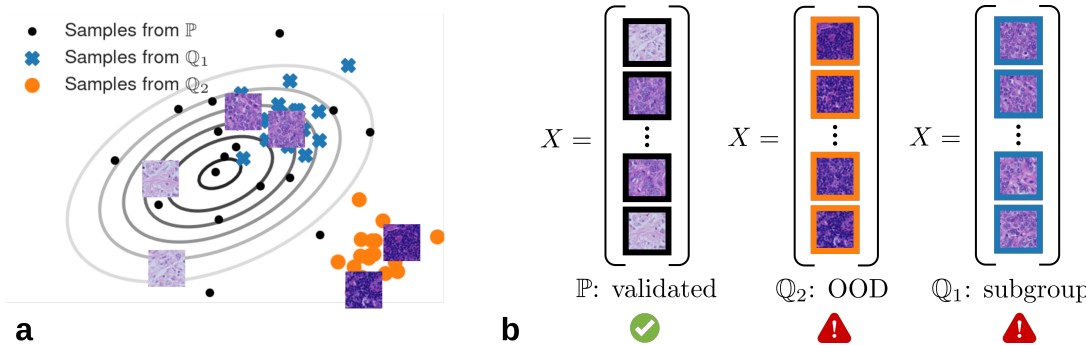

Figure 1: **a.** The contours depict the probability density $\mathbb{P}$. The blue and orange clusters are samples drawn from distributions $\mathbb{Q}_1$ and $\mathbb{Q}_2$, which exhibit a subgroup and OOD shift, respectively. **b.** Subgroup and OOD shifts render performance claims obtained from $\mathbb{P}$ invalid, but OOD detection would typically fail on examples drawn from subgroup $\mathbb{Q}_1$.

presence of tumour cells on histopathology samples may lead to inappropriate treatment and in the worst case, premature death. ML systems for healthcare in high risk settings are therefore subject to strict regulations to ensure their safety and effectiveness.

A key step in certifying ML systems is clinical validation, where the system's performance is assessed on validation data that are intended to be representative of the real data distribution encountered in the deployed system. After clinical validation, the system may be considered safe and approved for use in the intended setting by regulatory authorities. "Safe" in this context means that the benefits of using the ML system are considered to outweigh the risks associated with prediction errors. This risk-benefit analysis crucially hinges on realistic ML performance estimates: if the performance in the application setting falls short of the claimed performance estimated in the validation setting, the cumulative harm associated with prediction errors in all patients may exceed the level deemed acceptable.

One approach towards detecting changes in the application setting w.r.t. the validation setting is out-of-distribution (OOD) detection for individual examples. OOD detection is applied to each data point separately to identify whether it is unlikely to be drawn from the distribution the data was validated on, as this could lead to an unreliable prediction. There exists a large body of recent research on OOD detection (e.g. Liang et al. (2018); Hendrycks and Gimpel (2017); Daxberger et al. (2021); Erdil et al. (2021)). A comparison of OOD detection methods applied to medical images is provided in Jungo and Reyes (2019); Berger et al. (2021).

While the above-mentioned family of methods should be a crucial part of any safety-critical ML system, they are – as we will show – not suitable for detecting certain distribution shifts which can only be detected at a *population level*. In particular, conventional OOD detection methods cannot detect subgroup shifts, i.e. distribution shifts within the support of the original distribution (illustrated in Fig. 1). The blue and orange points depict a subgroup shift and an OOD shift, respectively (Fig. 1 **a**). Each individual point from the blue cluster is not atypical w.r.t. the source distribution, but they clearly are on a population

level (Fig. 1 **b**). While both types of shifts are problematic, OOD detection would typically fail on examples from the blue cluster, as also demonstrated experimentally in Sec. 3.2.

In reality, subgroup shifts can occur when subgroups are differently represented in data used to validate the algorithm compared to the application setting. For example, algorithms evaluated on population cohort data spanning all age groups may be predominantly applied in elderly people. As another example, algorithms for histopathology data analysis may be evaluated across multiple hospitals but applied predominantly in a subset of them, which could result in a overrepresentation of specific acquisition protocols, or screening devices used to obtain the data. While in both cases a shift could be detected by measuring and monitoring the age distribution or acquisition settings, crucial covariates characterising relevant subgroup attributes are often unmeasured or unidentified, and performance across subgroups may vary vary distinctly (Oakden-Rayner et al., 2020). When any distribution shifts (including subgroup shifts) occur, this may therefore impact the actual performance in the real application setting, or will at least render original performance claims invalid. Both scenarios pose serious problems for safely deploying machine learning systems.

We therefore aim to detect distribution shifts at a population level, with a specific focus on subgroup shift otherwise not detectable. We formulate shift detection in the framework of statistical hypothesis testing with a null hypothesis that two samples (i.e. sets of data points) are drawn from the same distribution. Recently developed hypothesis tests have reached meaningful statistical power on high-dimensional data for some types of distribution shift (Liu et al., 2020; Rabanser et al., 2019). As this problem has so far not been explored in a medical imaging setting, in this paper we focus on the following key contributions:

1. We demonstrate the fundamental limitations of classical OOD detection methods for detecting subgroup shifts, demonstrating the need for population-level shift detection.

2. We establish a baseline for subgroup shift detection on toy data as well as histopathology data. In particular, we demonstrate reliable subgroup shift detection with deep-kernel methods based on (Liu et al., 2020).

## 2. Methods for Distribution Shift Detection

We first introduce the exemplar state-of-the-art method (Sec. 2.1) we use to demonstrate that OOD detection fails at detecting subgroup shifts. We then revisit the framework of statistical hypothesis testing (Sec. 2.2) and introduce the population-level shift detection methods recently proposed by Liu et al. (2020) and Rabanser et al. (2019). As far as we are aware, these are the only existing methods potentially suitable for detecting subgroup shifts in medical images, and will form the basis for our baseline experiments.

### 2.1. Classical OOD detection

We use ODIN (Liang et al., 2018) for individual OOD detection, as it is widely used and performed best in a recent comparison of confidence-based OOD detection methods (Berger et al., 2021). Let us assume two probability distributions $\mathbb{P}, \mathbb{Q}$ over $\mathcal{X}$, where in our case, $\mathcal{X} = \mathbb{R}^n$ is the space of $n$-dimensional images. ODIN relies on a task classifier trained in $\mathbb{P}$, and uses the confidence of the predicted class (i.e. the maximum softmax output) to

predict whether a data point belongs to $\mathbb{P}$ or $\mathbb{Q}$ as in Hendrycks and Gimpel (2017). To improve detection, the input data is perturbed such that the softmax output for the true label increases, and temperature scaling is applied in the softmax layer (Liang et al., 2018). The temperature $\tau$ and perturbation magnitude $\epsilon$ are hyperparameters (see App. A).

## 2.2. Hypothesis Testing for Distribution-Shift Detection

Following Casella and Berger (2002) and Gretton et al. (2012), we formulate the problem of subgroup shift detection as a hypothesis test with null hypothesis $H_0 : \mathbb{P} = \mathbb{Q}$ and alternative $H_1 : \mathbb{P} \neq \mathbb{Q}$. In general, the hypotheses make a statement about a population parameter (e.g. difference in population mean for a two-sample $t$-test), and the test statistic $t(X, Y)$ is the corresponding estimate from the samples $X = \{x_i\}_{i=0}^m \overset{iid}{\sim} \mathbb{P}$ and $Y = \{y_i\}_{i=0}^m \overset{iid}{\sim} \mathbb{Q}$ (e.g. difference in sample means). $H_0$ is rejected for some rejection region of $t$. The *significance level $\alpha$* or *Type I error* denotes the probability that $H_0$ is rejected even if it is true. Typically, the rejection region is selected at a specific significance level, e.g. $\alpha = 0.05$. The *test power* denotes the probability that $H_0$ is correctly rejected if $H_1$ is true.

**Deep Kernel Tests (MMD-D)** Two-sample kernel tests (Gretton et al., 2012) use the Maximum Mean Discrepancy (MMD) on a Reproducing Kernel Hilbert Space (RKHS) as a test statistic.

**Definition 1** *(Gretton et al., 2012) Let $\mathcal{H}_k$ be a RKHS with kernel $k : \mathcal{X} \times \mathcal{X} \to \mathbb{R}$. The MMD is defined as*

$$MMD[\mathcal{H}_k, \mathbb{P}, \mathbb{Q}] = \sup_{f \in \mathcal{H}_k, \|f\|_{\mathcal{H}_k} \leq 1} \left( \mathbb{E}_{x \sim \mathbb{P}}[f(x)] - \mathbb{E}_{y \sim \mathbb{Q}}[f(y)] \right). \tag{1}$$

*For characteristic kernels $k$, the MMD is a metric and $MMD[\mathcal{H}_k, \mathbb{P}, \mathbb{Q}] = 0$ iff $\mathbb{P} = \mathbb{Q}$.*

The metric property in the above definition implies that an estimator $\widehat{\text{MMD}}(X, Y)$ is an appropriate test statistic for testing whether $\mathbb{P} = \mathbb{Q}$. An unbiased estimator for the MMD is (Gretton et al., 2012):

$$\widehat{\text{MMD}}(X, Y; k) = \frac{1}{m(m-1)} \sum_{i \neq j} H_{ij} \quad , \tag{2}$$

$$H_{ij} = k(x_i, x_j) + k(y_i, y_j) - k(x_i, y_j) - k(y_i, x_j) \tag{3}$$

For a given kernel (e.g. a Gaussian kernel), $\widehat{\text{MMD}}$ can be calculated on samples $X, Y$ and a permutation test can be used to determine whether $H_0 : \mathbb{P} = \mathbb{Q}$ can be rejected. The choice of the kernel $k$ affects test power in finite sample sizes and developing suitable kernels is an active area of research (e.g. Sutherland et al. (2017); Liu et al. (2020). We follow Liu et al. (2020) and parameterise the kernel $k_\theta$ with a neural network (see Appendix B for exact architecture). The kernel parameters are optimised on training data from $\mathbb{P}$ and $\mathbb{Q}$:

$$\theta = \arg \max_\theta \widehat{\text{MMD}}(X, Y; k_\theta) \tag{4}$$

It is important to note this diverges from Liu et al. (2020), where the objective function also incorporated knowledge on the asymptotic distribution of $\widehat{\text{MMD}}$ under $H_1$. We did not find this beneficial (see App. C for further analysis).

**Mass-Univariate Kolmogorov-Smirnov Tests on Task Predictions (MUKS)** Alternatively, Rabanser et al. (2019) propose hypothesis tests for domain shift detection that operate on low-dimensional representations of the original image space $\mathcal{X}$. They use a black-box classifier $g : \mathcal{X} \to \mathbb{R}^C$ trained on data from $\mathbb{P}$ as a dimensionality reduction technique, where $s = g(x) \in \mathbb{R}^C$ is a softmax prediction for $x$ with $C$ classes. A mass-univariate Kolmogorov-Smirnov (MUKS) test is then applied to the individual softmax predictions of samples $X$ and $Y$, yielding $C$ $p$-values. Rabanser et al. (2019) perform Bonferroni correction for multiple comparisons, i.e. $H_0$ is rejected if $p \leq \alpha/C$ for any of the $C$ comparisons.

## 3. Experiments and Results

We will first show that state-of-the-art OOD detection fails to detect distribution shifts in subgroups. Subsequently, we examine the shift detection capabilities of approaches based on hypothesis testing in a subgroup shift setting, evaluated on toy data and real-world histopathology images. We made the code available[1] used public datasets.

### 3.1. Data

**MNIST** As a toy example, we used the MNIST dataset (Lecun et al., 1998) with the official training and test splits (60'000 / 10'000 images). We held out 10'000 images from the training split for hyperparameter and model selection. Subgroup distribution shifts were modelled by artificially adjusting the prevalence of the digit 5 in $\mathbb{P}$ or $\mathbb{Q}$.

**Histopathology Images** We used the fully annotated Camelyon17 challenge data (Bándi et al., 2019) to demonstrate our results on real-world medical images. The dataset consists of 50 whole slide images (WSI) of H&E stained lymph node biopsies acquired across different hospitals. Due to differences in sample preparation and staining protocols, WSI typically vary severely across sites. We used a patch-based version of Camelyon17 that was recently provided by the WILDS domain generalisation benchmark (Koh et al., 2021) to study tumor detection across different hospitals. We split the data into training (284'219), validation (70'219) and test patches (100'820) while making sure no data from the test slides were used in the training and validation folds. Tumour prevalence was approximately 50% in all folds. Subgroup shifts were modelled by increasing the proportion of data from hospital 3. For this subgroup, shift detection is particularly important because of an observed drop in classification performance (Sec. 3.3.2)

### 3.2. Experiment 1: Limitations of Individual OOD Detection

#### 3.2.1. EXPERIMENTAL SETUP

To illustrate the inherent limitations of OOD detection methods for subgroup shifts, we applied the state-of-the-art OOD method ODIN (Liang et al., 2018) to a subgroup shift and OOD shift setting on MNIST. For the subgroup shift, we used MNIST as $\mathbb{P}$ (MNIST-all), and MNIST with exclusively 5's as $\mathbb{Q}$ (MNIST-5). For the OOD shift, we used MNIST *excluding* digit 5 as $\mathbb{P}$ (MNIST-no-5), and MNIST-5 as $\mathbb{Q}$ as before. For both settings, we trained a ResNet (He et al., 2016) as a digit classifier on the training fold of $\mathbb{P}$. For

---

1. https://github.com/lmkoch/subgroup-shift-detection

Table 1: Detection performance of ODIN (Liang et al., 2018) on MNIST, applied to OOD and subgroup shifts. Performance is reported using AUC (higher is better), FPR (lower is better) and detection rate (1-FPR).

| $\mathbb{P}$ (in) | $\mathbb{Q}$ (out) | Shift type | AUC | FPR (95% TPR) | Detection rate (95% TPR) |
|---|---|---|---|---|---|
| MNIST-no-5 | MNIST-5 | OOD | 0.96 | 0.20 | 0.80 |
| MNIST-all | MNIST-5 | subgroup | 0.44 | 0.96 | 0.04 |

the OOD shift setting, the digit classifier was trained on nine digits (excluding 5). Both classifiers reached an accuracy of 0.98 for the respective task on the validation split. The hyperparameters for temperature and perturbation magnitude were chosen based on a grid search in the validation set (see App. A).

As is common, the performance of ODIN was evaluated with the area under the ROC curve (AUC) as well as the false positive rate (FPR) at a true positive rate of 95%. For calculating these measures, $\mathbb{P}$ was assumed the positive class. The shift detection rate (1-FPR) was added for ease of interpretation, denoting the proportion of examples from $\mathbb{Q}$ that were correctly identified.

### 3.2.2. RESULTS

The detection performance of ODIN for the OOD setting was markedly better than for subgroup shifts (Table 1). According to these measures, ODIN accurately detected OOD shifts (AUC 0.96, detection rate 0.80), but showed close to chance performance for subgroup shifts (AUC 0.44, detection rate 0.04). This experiment shows that OOD detection methods cannot be used to detect subgroup shifts. These findings could also be confirmed with evaluation on other shift settings and additional OOD detection methods (see App. D).

### 3.3. Experiment 2: Population-Level Subgroup Shift Detection

#### 3.3.1. EXPERIMENTAL SETUP

We examined the population-level shift detection capabilities of the approaches outlined in Sec. 2.2 (MMD-D) and Sec. 2.2 (MUKS) in the subgroup shift setting. We studied subgroup shifts with over-representations of subgroups (MNIST: digits 5, Camelyon: hospital 3) of factor $w = \{1, 5, 10, 100\}$ in the target distribution $\mathbb{Q}$ w.r.t. source distribution $\mathbb{P}$. The configuration $w = 1$ denotes the case where there was no over-representation, i.e. $\mathbb{P} = \mathbb{Q}$.

**Deep kernel test (MMD-D)** For each shift configuration, we trained a deep kernel to optimise the power of the statistical hypothesis test based on MMD-D. The kernel architecture was chosen as in (Liu et al., 2020). The training used minibatches of size 64 from the training fold of both $\mathbb{P}$ and $\mathbb{Q}$. We used the Adam optimiser (Kingma and Ba, 2015) with a learning rate of $10^{-5}$.

**Mass-univariate KS test (MUKS)** The MUKS hypothesis test for shift detection relies on a task classifier in the source distribution $\mathbb{P}$. For MNIST, we re-used the classifier from

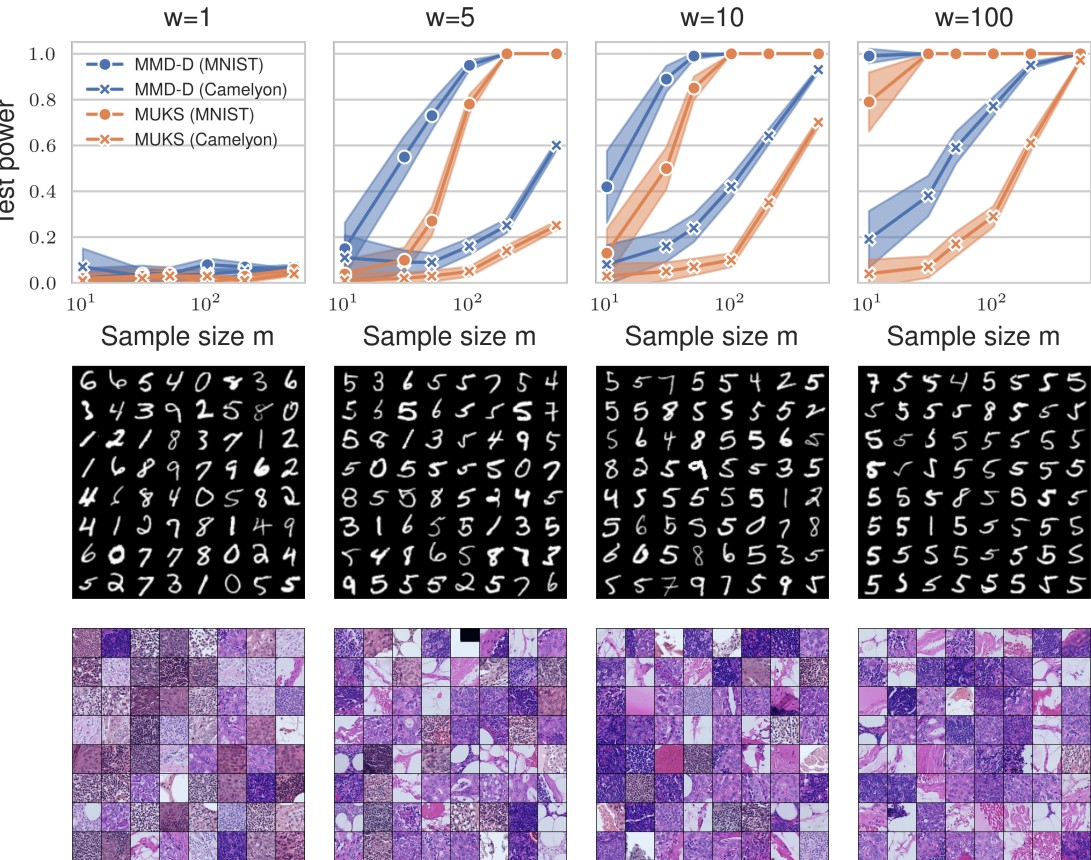

Figure 2: The top row shows test power (shaded area depicts standard error) of MMD-D and MUKS on MNIST (●) and Camelyon17 (✖) for varying degrees of subgroup shift, from no shift ($w = 1$, left) to strong shift ($w = 100$, right). The bottom rows show example images from different subgroup shifts $\mathbb{Q}$ for MNIST and Camelyon17.

Sec. 3.2. For Camelyon17 data, we trained a DenseNet (Huang et al., 2017) using the training regime proposed in Koh et al. (2021). We applied the MUKS test to the softmax outputs of classifier predictions on test data from $\mathbb{P}$ and $\mathbb{Q}$ as described in Sec. 2.2.

**Performance evaluation** As in Liu et al. (2020); Rabanser et al. (2019), we assessed the shift detection performance through the test power, which we calculated as the shift detection rate, i.e. the proportion of rejected null hypotheses $H_0 : \mathbb{P} = \mathbb{Q}$ in repeated experiments (100 repetitions at significance level $\alpha = 0.05$). For each repetition, we drew samples of size $m \in \{10, 30, 50, 100, 200, 500\}$ from the test folds of $\mathbb{P}$ and $\mathbb{Q}$. We calculated the empirical Type I error similarly by repeatedly drawing two independent samples of $\mathbb{P}$.

### 3.3.2. RESULTS

For strong subgroup shifts in MNIST ($w = 100$, i.e. almost exclusively digit 5 present in $\mathbb{Q}$) both MMD-D and MUKS yielded perfect test power with a sample size $m \geq 30$ (top right in

Fig. 2). In the same setting on histopathology images, both approaches approached perfect test power for a sample size $m = 500$. With more subtle subgroup shifts ($w = 5, 10$, middle figures), the test power decreased gradually when evaluated for smaller sample sizes, but on MNIST reached perfect test power for $m = 200$ (subgroup shift $w = 5$) and $m = 100$ (subgroup shift $w = 10$) for both MMD-D and MUKS (see App. E for some supplementary results). The experiments on histopathology data yielded reduced performance compared to MNIST, likely because the changes were more subtle (Fig. 2 bottom rows), but the overall trends were consistent. Importantly, both approaches correctly failed to detect a shift when no actual subgroup shift was present (Fig. 2, top, $w = 1$, i.e. $\mathbb{P} = \mathbb{Q}$). In this case, as expected, the rate at which $H_0$ was rejected was approximately equal to the chosen significance level $\alpha = 0.05$. The observed Type I error for all tests was also consistent with $\alpha = 0.05$: the mean (SD) across all tests was 0.05 (0.02) for MMD-D and 0.03 (0.02) for MUKS on both datasets (the latter is expected to be lower than the desired 0.05 due to the overly conservative Bonferroni correction).

In summary, MMD-D performed consistently better than MUKS, but overall, both methods were able to effectively detect subgroup shifts. This is in clear contrast to the detection performance obtained with classical OOD detection as demonstrated in Sec. 3.2.

To demonstrate the impact of subgroup shifts, we compared Camelyon17 classification performance in all hospitals (acc $= 0.79$) to performance on the subgroup (hospital 3, acc $= 0.73$). This was possible as we had access to both task labels and subgroup attributes. This shows that a subgroup shift towards hospital 3, as studied here, would have a negative impact on classification performance in the application setting.

## 4. Discussion and Conclusion

We demonstrate in this paper that subgroup shifts escape OOD detection completely even for very strong shifts. While this is not a surprising result given that OOD detection was not designed to operate in this setting, it has implications for the safe application of machine learning in safety-critical settings. For the studied subgroup shift, the results can be contrasted directly with an effective detection using either of the population-level approaches. In addition to improving robustness to distributions shifts, we argue that population-level distribution shift detection should be used alongside OOD detection to facilitate trust in the real-world performance of ML systems.

Comparing the hypothesis testing approaches, MMD-D clearly and consistently outperformed MUKS. This is likely because MMD-D was trained with access to data from the application setting. When availability of real data is limited, this may motivate the use of MUKS. However, as MMD-D does not requires any labelled examples, data availability may not be a crucial limitation.

As subgroup shift detection has, to our knowledge, not been explored on medical data, we focused this paper on formulating and motivating the problem setting and establishing a first baseline on real-world data. However, our work suggests several avenues for further research. We expect that both hypotheses tests studied in this paper can be improved further through more expressive kernel architectures for MMD-D, and more powerful task classifiers for MUKS. To broaden the clinical impact, we will investigate other types of clinically relevant population shifts, e.g. in protected subgroups based on gender or ethnicity.

## Acknowledgments

This work was supported by the German Ministry of Science and Education (BMBF, 01IS18039A) and the German Science Foundation (BE5601/4-1 and EXC 2064 ML, project number 390727645).

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

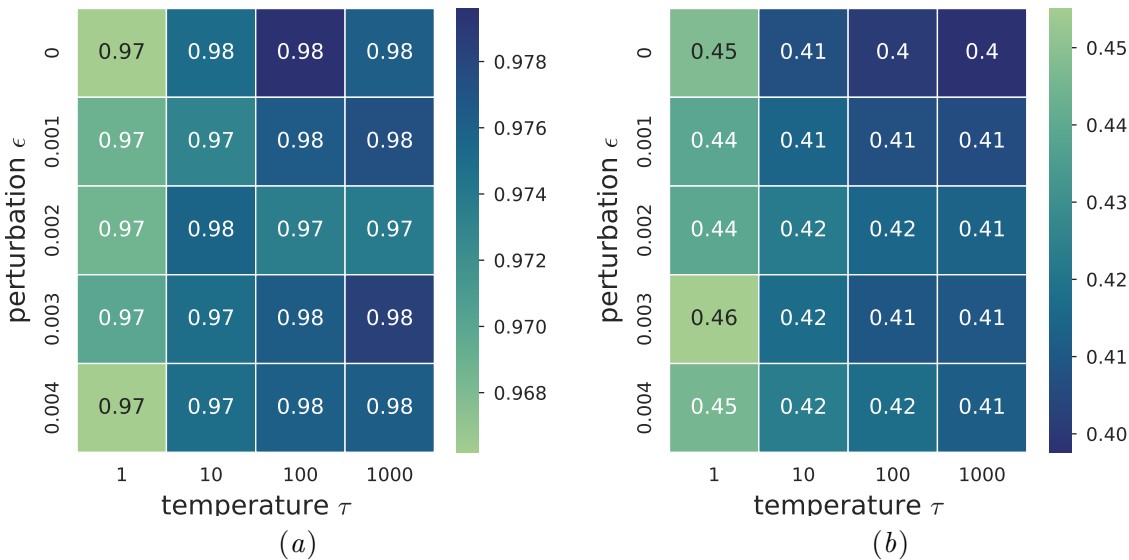

Figure 3: This figure shows subgroup shift detection performance on the MNIST validation set for all hyperparameter choices $\tau, \epsilon$. (a) shows FPR@TPR95 (lower is better), and (b) shows ROC AUC (higher is better).

## Appendix A. Temperature Scaling and Input Perturbation for ODIN

The softmax predictions after temperature scaling can be calculated as (Liang et al., 2018):

$$g^{(c)}(x; \tau) = \frac{\exp\left(f^c(x)/\tau\right)}{\sum_{j=1}^{C} \exp\left(f^j(x)/\tau\right)} \quad , \tag{5}$$

where $f(x)$ denotes a logit prediction. In addition, the inputs were perturbed to increase the softmax output for the predicted class label $\hat{c}$ (Liang et al., 2018):

$$\tilde{x} = x - \epsilon \text{sign}\left(-\nabla_x \log g^{(\hat{c})}(x; \tau)\right) \quad . \tag{6}$$

Figures 3 and 4 show the grid search results obtained for the subgroup shift and OOD shift settings, respectively, using the exploration range $\tau \in \{1, 10, 100, 1000\}, \epsilon \in \{0, 0.001, 0.002, 0.003, 0.004\}$. The hyperparameters were selected to minimise the false positive rate (FPR@TPR95), resulting in $\tau = 1, \epsilon = 0.0$ for the subgroup shift setting and $\tau = 1, \epsilon = 0.002$ for the OOD shift setting.

## Appendix B. Kernel Architecture Deep Kernel Test

In this paper, we used the kernel proposed in Liu et al. (2019):

$$k_\theta(x, y) = ((1 - \delta)f(\phi_\theta(x), \phi_\theta(y)) + \delta) g(x, y) \quad , \tag{7}$$

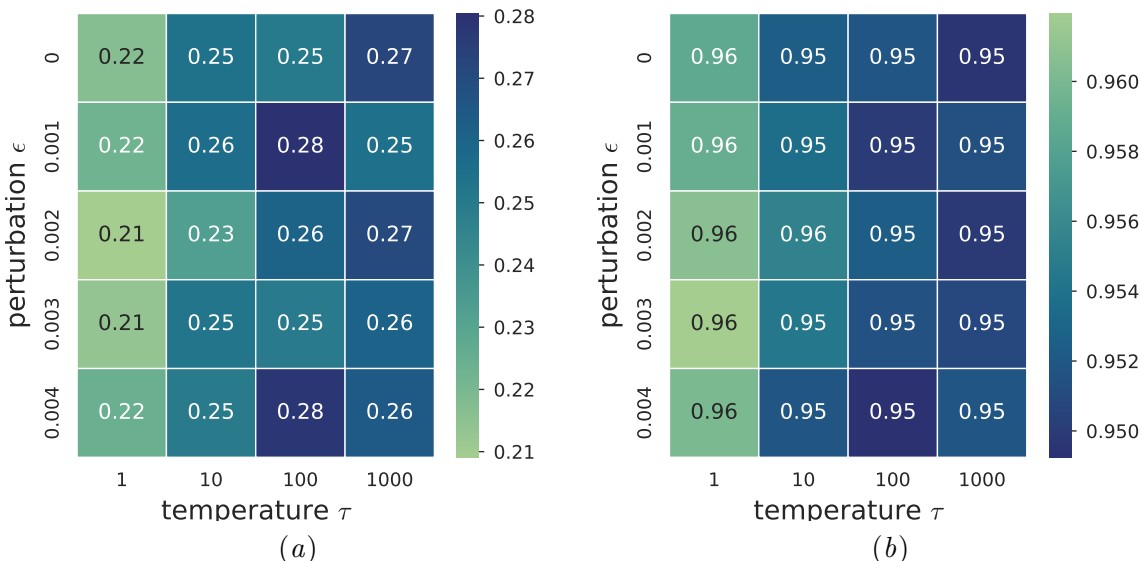

Figure 4: This figure shows OOD shift detection performance on the MNIST validation set for all hyperparameter choices $\tau, \epsilon$. (a) shows FPR@TPR95 (lower is better), and (b) shows ROC AUC (higher is better).

where $f$ and $g$ are Gaussian kernels taking pairs of deep feature vectors $\phi_\theta(x), \phi_\theta(y)$ and original data $x, y$ as inputs, respectively. Both Gaussian kernels have a lengthscale parameter that is also optimised with Eq. 9 along with $\theta$. The architecture of the feature extractor $\phi_\theta$ consists of four convolutional layers followed by a linear layer as used in the DCGAN discriminator (Radford et al., 2016).

## Appendix C. Estimators and Architecture for Optimal MMD Kernel Tests

Sutherland et al. (2017) proposed a mechanism for choosing a kernel $k_\theta$ that maximises the test power by exploiting knowledge of the variance $\sigma_{H_1}$ of the asymptotic sampling distribution of the test statistic under $H_1$. An estimator $\hat{\sigma}^2_{H_1, \lambda}$ for the variance is (Liu et al., 2020)

$$\widehat{\sigma^2_\lambda}(X, Y; k_\theta) = \frac{4}{n^3} \sum_{i=1}^{m} \left( \sum_{i=1}^{m} H_{ij} \right)^2 - \frac{4}{n^4} \left( \sum_{i=1}^{m} \sum_{i=1}^{m} H_{ij} \right)^2 + \lambda \quad . \tag{8}$$

The test power can then be optimised by finding kernel parameters that optimise the objective function

$$J_\lambda(X, Y; \theta) = \frac{\widehat{\mathrm{MMD}}(X, Y; k_\theta)}{\hat{\sigma}_{H_1, \lambda}(X, Y; k_\theta)} \quad . \tag{9}$$

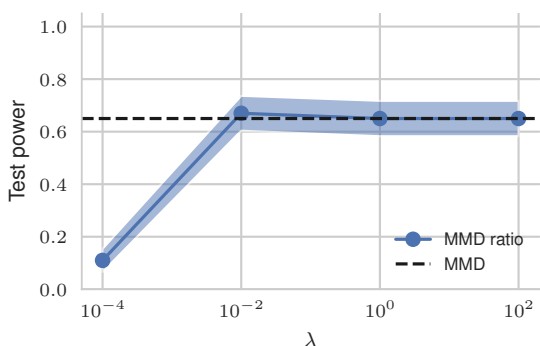

Figure 5: Test power for different choices of $\lambda$ is shown in blue. Overlaid is the test power obtained by maximising without considering the denominator of Eq. 9. Here, we assessed test power for samples of size $m = 50$.

Optimising test power requires a hyperparameter $\lambda$ which controls the contribution of the variance of of the test statistic under $H_1$. Examining various hyperparameter values, we surprisingly found that using the objective in Eq. 9 did not yield a benefit over maximising $\widehat{\mathrm{MMD}}(X, Y; k_\theta)$ only (Eq. 4). Results of a hyperparameter search are shown in Fig. 5. The experiment was carried out using MNIST to model $\mathbb{P}$, and MNIST with a 5-fold oversampling of digit 5 as $\mathbb{Q}$.

## Appendix D. Additional Results for Individual OOD Detection

Here, we show additional results for classical OOD detection in OOD and subgroup shift settings (see Table 2). Distribution shifts were modelled by changing the proportion of each individual digit in MNIST. In addition to ODIN, which was evaluated in Sec. 3.2, we show results on two other OOD detection methods: "Baseline" (Hendrycks and Gimpel, 2017) using the maximum probability output to distinguish domains, and "Laplace" (Daxberger et al., 2021), which is an approach based on Laplace approximations of the loss function and has been shown to be useful for OOD detection.

The results for the different digits were consistent with the results reported in Sec. 3.2, with classical OOD detection failing catastrophically on subgroup shift settings (bottom rows in Table 2) for all digits, and obtaining detection rates between 0.7-0.95 for OOD shift settings (top rows in Table 2).

## Appendix E. Additional Results for Population-Level Subgroup Shift Detection

Here, we expand the evaluation of population-level subgroup shift detection (Sec. 3.3) by modelling additional subgroup shifts in MNIST. Similarly to the additional experiments in App D), we modelled these shifts by changing the proportion of each individual digit in

Table 2: Detection performance of OOD detection methods on MNIST, applied to OOD (top rows) and subgroup shifts (bottom rows). Performance is reported using AUROC (higher is better) and detection rate (1-FPR, higher is better).

| $\mathbb{P}$ (in) | $\mathbb{Q}$ (out) | AUC | Detection rate (95% TPR) |
|---|---|---|---|
| | | Baseline / Laplace / ODIN | |
| MNIST-no-0 | MNIST-0 | 0.98 / 0.98 / 0.98 | 0.91 / 0.94 / 0.92 |
| MNIST-no-1 | MNIST-1 | 0.98 / 0.99 / 0.98 | 0.89 / 0.95 / 0.89 |
| MNIST-no-2 | MNIST-2 | 0.96 / 0.97 / 0.96 | 0.77 / 0.79 / 0.77 |
| MNIST-no-3 | MNIST-3 | 0.95 / 0.96 / 0.95 | 0.70 / 0.73 / 0.70 |
| MNIST-no-4 | MNIST-4 | 0.95 / 0.95 / 0.95 | 0.78 / 0.76 / 0.77 |
| MNIST-no-5 | MNIST-5 | 0.96 / 0.96 / 0.96 | 0.80 / 0.78 / 0.80 |
| MNIST-no-6 | MNIST-6 | 0.97 / 0.98 / 0.97 | 0.87 / 0.90 / 0.87 |
| MNIST-no-7 | MNIST-7 | 0.97 / 0.97 / 0.97 | 0.87 / 0.87 / 0.88 |
| MNIST-no-8 | MNIST-8 | 0.97 / 0.96 / 0.97 | 0.83 / 0.81 / 0.83 |
| MNIST-no-9 | MNIST-9 | 0.96 / 0.97 / 0.96 | 0.79 / 0.80 / 0.78 |
| MNIST-all | MNIST-0 | 0.32 / 0.34 / 0.32 | 0.02 / 0.02 / 0.02 |
| MNIST-all | MNIST-1 | 0.46 / 0.43 / 0.46 | 0.02 / 0.01 / 0.02 |
| MNIST-all | MNIST-2 | 0.52 / 0.51 / 0.52 | 0.04 / 0.03 / 0.04 |
| MNIST-all | MNIST-3 | 0.44 / 0.40 / 0.44 | 0.04 / 0.04 / 0.04 |
| MNIST-all | MNIST-4 | 0.47 / 0.46 / 0.47 | 0.05 / 0.05 / 0.05 |
| MNIST-all | MNIST-5 | 0.44 / 0.41 / 0.44 | 0.04 / 0.04 / 0.04 |
| MNIST-all | MNIST-6 | 0.47 / 0.60 / 0.47 | 0.05 / 0.06 / 0.05 |
| MNIST-all | MNIST-7 | 0.70 / 0.72 / 0.70 | 0.06 / 0.06 / 0.06 |
| MNIST-all | MNIST-8 | 0.51 / 0.47 / 0.51 | 0.06 / 0.07 / 0.06 |
| MNIST-all | MNIST-9 | 0.70 / 0.64 / 0.70 | 0.12 / 0.11 / 0.12 |

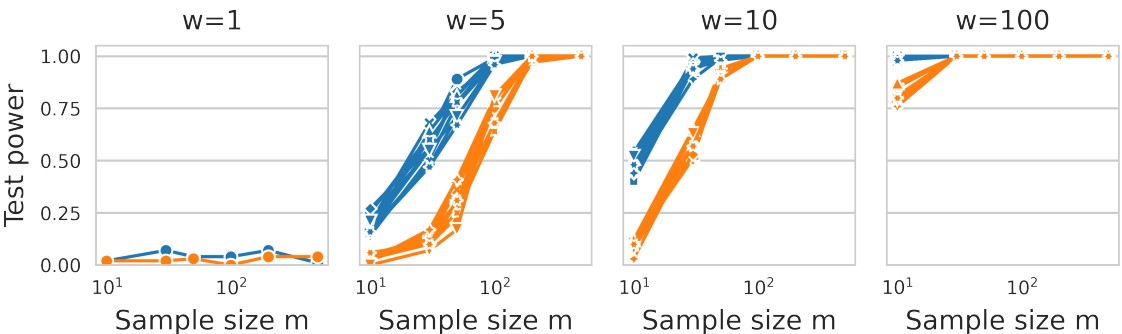

Figure 6: The panels shows test power of MMD-D (blue) and MUKS (orange) on MNIST for varying degrees of subgroup shift, from no shift ($w = 1$, left) to strong shift ($w = 100$, right). Different curves of the same color denote shifts w.r.t. different digits.

MNIST, leading to relatively consistent results (see Fig. 6) with some variability in test power for different digits.

