# OpenReview forum: "Hidden in Plain Sight: Subgroup Shifts Escape OOD Detection"
_MIDL.io/2022/Conference — MIDL 2022_

### Official Review · Reviewer_spWg · 2022-01-19

**Confidence:** 5
**Preliminary Rating:** 2

**Summary:**

The authors claim that current out-of-distribution detection algorithms failed in capturing the subgroup / population-level shifts in medical data. They examined the performance of classical OOD detection methods on a modified MNIST dataset. Then, they validated two statistical hypothesis testing methods, namely deep kernel test and mass-univariate KS test for subgroup shift detection on both the modified MNIST and tumor histopathology dataset. In the real dataset, the degree of subgroup shift is represented as the proportion of images acquired from an unseen hospital. And they found that the two methods can successfully detect the subgroup shift.

**Strengths:**

Subgroup shift / distribution shift is an interesting topic in medical imaging where deep learning algorithms are not robust to these shifts in real scenarios. The paper has some discussions on the importance of clinical validation for ML systems.

**Weaknesses:**

1. There are weaknesses in the experimental design. For example, the classical OOD detection method has been only tested on MNIST but not on the real histopathology dataset. Similarly, the evaluation metrics for the classical OOD detection and the two statistical hypothesis testing methods are different.
2. The subgroup shift presented in the paper is site-shift. This kind of shift is very obvious and very likely happens when we get data from multi-centers. Is it necessary to detect if there’s any shift between different centers? A more important issue is how to overcome the shift. For example, we can remove the site-shift by using harmonization methods or adjusting the prediction models using domain adaptation techniques.
3. As a validation / empirical study paper, only one real clinical dataset has been used for the experiments. Also there's a lack of diversity of the population-level shifts.


**Deanonymize Review:**

no

**Final Rating After The Rebuttal:**

4: Weak Accept

**Justification Of The Final Rating:**

The authors have addressed most of my concerns. This is an important research problem and the authors made their first attempt on addressing it. I'm looking forward to see their followup studies. I would like to upgrade my rating from 2 to 4.

**Paper Type:**

validation/application paper

**Questions To Address In The Rebuttal:**

1. Subgroup shift is not properly defined in the paper. In the histopathology experiments, shifts might not only come from population but also from acquisition protocols, or screening device changes.
2. The evaluation metrics between the classical OOD detection method and the population-level shift methods are different.
3. The dataset is on a relatively small scale and less diverse for a validation study.
4. I’m not convinced by the motivation of detecting the subgroup shift of data at least in the histopathology experiments where it’s obvious that data from different hospitals suffer from distribution shift due to site effects.
5. The paper is hard to follow in general, especially there’s lack of coherence between paragraphs.


**Special Issue:**

no

---

### Official Review · Reviewer_BuUg · 2022-01-23

**Confidence:** 5
**Preliminary Rating:** 1

**Summary:**

This paper discusses the challenge of subgroup shift which is where a deployed model is run on data which is considered similar enough to the training data but has low performance.

The paper also presents a method for detecting this type of shift using the MMD metric with a neural network as kernel.


**Strengths:**

The proposed method is interesting and has many parallels to contrastive and deep metric learning. It also seems similar to minibatch discrimination (section 3.2) in https://arxiv.org/abs/1606.03498 I think these works should also be discussed to frame the method well within the literature.


**Weaknesses:**

The problem that is being solved is not sufficiently motivated as a problem that should be solved. If a model's performance was fine on an example when deployed why is it an issue if that model is used on those sample data points again?

Only one baseline OoD method is used when claiming that existing OoD methods cannot detect this shift. Although it is not clear why any need to be evaluated because the result is defined to work as the samples are learned to be in distribution and there is no batch wise consideration. This experiment doesn't seem necessary to include in the paper or perform more of an exhaustive search of OoD methods.


**Deanonymize Review:**

no

**Final Rating After The Rebuttal:**

3: Borderline

**Justification Of The Final Rating:**

I still think that the question this research paper addresses is not something we should be working on and I am not convinced by this paper otherwise.

But the method presented in the paper does not appear wrong and the question is a voice in the public conversation about how to deal with OoD examples so I increased by score to 3.

My fear is that acceptance of this work stands to validate this research direction and further work will use the existence of this publication as justification of their work without critically thinking about why we are solving this problem.

**Paper Type:**

methodological development

**Questions To Address In The Rebuttal:**

The paper states "it has implications for the safe application of machine learning in safety-critical settings". Please enumerate these safety challenges. Please address the argument that if a system was safe when deployed why is it now not safe when processing only a subset of the same data.

**Special Issue:**

no

---

### Meta-Review · Area_Chair_DFhi · 2022-02-16

**Recommendation:** Accept (Poster)
**Confidence:** 5

**Metareview:**

The paper addresses out of distribution detection on a subgroup level.  The discussion of the paper has been divided. Two reviewers agreed on the importance on the topic, and one reviewer found it outside of topics that should be studied, leading to an engaging discussion. Similarly to the authors and two of the reviewers, I find the topic worth studying. We should of course also critically think about what we are studying as a field, but in this instance I find that 1) the authors have done so, and 2) this is not an appropriate place to have this more general discussion, or to reject this specific paper while others in the field are studying this topic. I will take this point up further to discuss with the program chairs.

Next to the motivation of the topic, all reviewers suggested additional explanations and experiments, which the authors have successfully addressed, leading to improved scores for the paper for all reviewers. I would like to thank everyone for engaging in the discussion and am happy to recommend acceptance of the paper.

---

### Decision · Program_Chairs · 2022-02-28

Accept